# Underwater Ambiguity Elimination Method Based on Co-Prime Sensor Array

**DOI:** 10.3390/s20216058

**Published:** 2020-10-24

**Authors:** Tian Lan, Yilin Wang, Longhao Qiu

**Affiliations:** 1Acoustic Science and Technology Laboratory, Harbin Engineering University, Harbin 150001, China; lantianbluesky@hrbeu.edu.cn (T.L.); qiulonghao@hrbeu.edu.cn (L.Q.); 2Key Laboratory of Marine Information Acquisition and Security (Harbin Engineering University), Ministry of Industry and Information Technology, Harbin 150001, China; 3College of Underwater Acoustic Engineering, Harbin Engineering University, Harbin 150001, China; 4Qingdao Haina Underwater Information Technology Co., Ltd., Qingdao 266500, China

**Keywords:** DOA estimation, co-prime array, ambiguity elimination

## Abstract

Recently, the direction of arrival estimation with co-prime arrays has gradually been applied in underwater scenarios because of its significant advantages over traditional uniform linear arrays. Despite the advantages of co-prime arrays, the spatial spectra obtained directly from conventional beamforming can be degraded by grating lobes due to the sparse spatial sampling in passive sensing applications, which will seriously deteriorate the estimation performance. In this paper, capon beamforming is applied to a co-prime sensor array as a pretreatment before high-resolution direction of arrival (DOA) estimation methods. The amplitudes extracted from the beam-domain outputs of two subarrays and the phases extracted from the cross-spectrum of the spatial spectrum are exploited to suppress the spurious peaks in beam patterns and eliminate ambiguities. Consequently, interference can be further mitigated, and the performance of high-resolution DOA methods will be guaranteed. Simulations show that the method proposed can improve the reliability and accuracy of DOA estimation with great value in practice.

## 1. Introduction

Direction of arrival (DOA) estimation is an important array signal processing technique that finds broad applications in underwater passive sonar systems. An important goal for DOA estimation is to be able to locate closely spaced sources in the presence of considerable noise. Utilizing this technique, passive sonar systems can detect, localize, and track underwater acoustic sources by monitoring with an array with spatially separated sensors. DOA estimation has been an active research area during the last two decades. Conventional techniques for DOA estimation, such as beamforming, have been widely applied in the field of passive sonar. However, they can only provide limited angular resolution. Well-known methods to improve angular resolution are implementation of longer arrays or super-resolution subspace-based methods. Unfortunately, the applicability of both techniques is constrained when applied in a complex underwater environment. Firstly, the implementation of long towed linear arrays (TLA) dedicated to platforms such as unmanned underwater vehicles (UUVs) is frequently impossible because of structure constraints (e.g., heavy cables). Secondly, in realistic underwater environments, the received signals are temporally correlated as a result of multi-path arrivals. Under these conditions, the performance of subspace-based methods, such as multiple signal classification (MUSIC), will significantly degrade. Therefore, how to approach the problems mentioned above has attracted much interest in recent years [1,2].

Traditional beamforming and spatial spectrum estimation algorithms are mostly based on uniform linear half-wavelength arrays (ULAs). For conventional ULAs, it is necessary to increase the number of sensors in order to obtain a high resolution coupled with the aperture of the array. This leads to higher critical hardware cost and more difficulty in array design [3,4]. Sparse arrays have considerable advantages over the conventional ULAs. When the number of sensors is the same, sparse arrays will occupy more space, which means longer array geometry, larger aperture, and increased degrees of freedom (DOF). Because of this, higher accuracy and resolution of DOA estimation can be achieved [5]. When the aperture is fixed, sparse arrays require fewer physical sensors and related electronics, and consequently reduce the physical array cost. In addition, due to the expansion of the inter-spacing of sensors, mutual coupling, whose magnitude is inversely proportional to the inter-element distance, will be reduced, and the performance of DOA estimation may suffer less from coupling nonideality [6]. In general, a longer distance means less coupling, but in practice it also depends on the platforms, materials, etc. With these advantages, sparse arrays are being gradually and widely applied to underwater applications.

The existing sparse array structures include the Wichmann Array [7], Minimum Redundancy Array (MRA) [8], Minimum Hole Array (MHA) [9], Sparse Ruler Array (SRA) [10], Nested Array (NA) [11], Co-prime Array (CA) [12,13], and so on. Among these arrays, the MRAs and MHAs have no closed-form expressions for their array geometries. NAs [14,15] and CAs [16,17] have been proposed and recently been widely studied. These two types of sparse arrays can provide closed-form expression for finding the array geometry, and they have obvious advantages on the design of array configurations utilizing a difference co-array (virtual array), which is crucial for sparse array modeling. Combined with relevant signal processing algorithms, sparse arrays can achieve the desired effect of detection and localization [18,19]. Meanwhile, the array combination technology based on sparse arrays, which is different from the co-array-like models, has been proposed and studied recently [20].

It is widely acknowledged that because of the sparse spatial sampling in sparse arrays, grating lobes will appear and deteriorate the estimation performance. On basis of recent CA research, many solutions have been proposed to avoid the ambiguities caused by grating lobes. The solutions include processing the beam-domain output of two subarrays by a product processor or minimal processor [21,22], eliminating spurious peaks with fourth-order cumulants [23], ad hoc methods such as DECOM [24], and so on. Many applications, such as radar and underwater surveillance using hydrophones, can take advantage of co-prime arrays. In the remote sensing field, the product and minimal processor are widely applied [25,26]. In phased array and through-the-wall radar fields, the grating lobes suppression algorithms have been actively studied [27,28]. Although the relevant state-of-the-art works can reduce ambiguities in certain applications, there are some cases that should be considered. Assume that a CA with several subarrays is applied to a multi-targets location scenario. When DOA estimation such as conventional beamforming or MUSIC is carried out to each subarray, the grating lobes will be generated due to longer inter-element spacing than λ/2. When the grating lobes of different subarrays overlap after product or minimal processing, some spurious peaks will appear and cannot be eliminated. Obviously, the spurious peaks lead to ambiguities and affect the performance of DOA estimation. In order to solve this problem, this paper proposes a corresponding processing method. The method is based on the co-prime properties of CA and the information extracted from subarrays, which can effectively identify spurious peaks and reduce ambiguities, especially when there are many targets. The proposed method can be further applied to many fields like underwater surveillance with high practical value.

This paper is organized as follows: Section 2 introduces the array geometry, signal model, and beamforming method. Section 3 presents the grating lobes suppression for single targets and multi-targets with the proposed processing flow. Simulation and discussion are shown in Section 4, and Section 5 concludes the paper.

## 2. Array Structure, Signal Model, and Beamforming

In this paper, the CA structure is composed of two co-prime subarrays. The positions of the 1st sensor in the two subarrays coincide, and the sensors in each subarray are arranged evenly and equidistantly. The inner spacing of sensors in the two subarrays is co-prime. As shown in Figure 1, the co-prime pair of the two subarrays is (M,N), where *M* and *N* are both positive integers (M,N≥2), and the array expansion factor is α. In subarray one, the number of sensors is αN+1 and the inner spacing is Md. In subarray two, the number of sensors is αM+1 and the inner spacing is Nd. d=λ/2 is the half-wavelength of the impinging signal. Conventional beamforming (CBF) is employed as the signal processing method in this paper for conciseness, which can be further upgraded to MVDR methods or other robust beamforming algorithms. The scanning range is θ∈(−π/2,π/2).

Assume that all the targets are in the far-field and the sources are all narrowband signals. As is shown in Figure 2, the directions of k=1,…K targets are $\theta_{k}$, k=1,…K. Suppose $s_{k}\left(t\right)$ is the signal model of the *k*th target, then
(1)$$s_{k}\left(t\right)=A_{k}\left(t\right)e^{-j(2\pi ct/\lambda_{k}\left(t\right)+\varphi_{k}\left(t\right))}\;k=1,2,\ldots ,K$$

In Equation (Equation 1), $A_{k}\left(t\right)$ is the amplitude of $s_{k}\left(t\right)$, $\varphi_{k}\left(t\right)$ is the phase of $s_{k}\left(t\right)$, and *c* is the velocity of sound. In an underwater sensing environment, all the parameters of the signals $s_{k}\left(t\right)$ are assumed to be random in this signal model. As a matter of fact, the original source signal parameters are basically different from each other. The amplitudes at the receiving point are related to the propagation distance from the sources, so $A_{k}\left(t\right)$ is random. When narrowband coherent sources exist or the multi-path effects appear, the wavelength of received signals is influenced, and then $\lambda_{k}\left(t\right)$ is random. For the influence of the environment, the phase $\varphi_{k}\left(t\right)$ of the received signals is also random, which is in the range of (0,2π) [29,30]. For the amplitude and phase of the far field signal at time *t* and time t+τ, we have $A_{k}\left(t+\tau \right)\approx A_{k}\left(t\right)$ and $\varphi_{k}\left(t+\tau \right)\approx \varphi_{k}\left(t\right)$. Then the relationship between $s_{k}\left(t+\tau \right)$ and $s_{k}\left(t\right)$ is
(2)$$s_{k}\left(t+\tau \right)\approx A_{k}\left(t\right)e^{-j(2\pi ct/\lambda +\varphi_{k}\left(t\right))}e^{-j2\pi c\tau /\lambda }=s_{k}\left(t\right)e^{-j2\pi c\tau /\lambda }$$

In Equation (Equation 2), $s_{k}\left(t\right)$ is the *k*th signal received at time *t*, and the the impinging signal angle is $\theta_{k}$, k=1,…K. Then the received signal $x_{n}\left(t\right)$ and $x_{m}\left(t\right)$ can be presented as Equation (Equation 3),
(3)$$\left\{\begin{array}{c} x_{n}\left(t\right)=\sum_{k=1}^{K}s_{k}\left(t-\tau_{nk}\right)+n_{n}\left(t\right)\\ x_{m}\left(t\right)=\sum_{k=1}^{K}s_{k}\left(t-\tau_{mk}\right)+n_{m}\left(t\right) \end{array}\right.$$

In Equation (Equation 3), $n_{n}\left(t\right)$ is the additive white Gaussian noise with unknown variance at the *n*th receiver sensor of subarray one, n=1,…,αN+1 and $n_{m}\left(t\right)$ is the noise at the *m*th receiver sensor of subarray two, m=1,…,αM+1. $\tau_{nk}$ is the relative time lag for the *k*th signal at the *n*th sensor in subarray one, and $\tau_{mk}$ is the relative time lag for the *k*th signal at the *m*th sensor in subarray two. With Equation (Equation 2), Equation (Equation 3) can be expressed as
(4)$$\left\{\begin{array}{c} x_{n}\left(t\right)=\sum_{k=1}^{K}s_{k}\left(t\right)e^{-j2\pi nd\sin\theta_{k}/\lambda }+n_{n}\left(t\right)\\ x_{m}\left(t\right)=\sum_{k=1}^{K}s_{k}\left(t\right)e^{-j2\pi md\sin\theta_{k}/\lambda }+n_{m}\left(t\right) \end{array}\right.$$

In Equation (Equation 4), $\theta_{k}$ is the direction of the *k*th signal, k=1,…,K. Then, the received signal vectors of two subarrays can be expressed as
(5)$$\left\{\begin{array}{c} x_{1}\left(t\right)=A_{1}\left(\theta \right)s\left(t\right)+n_{1}\left(t\right)\\ x_{2}\left(t\right)=A_{2}\left(\theta \right)s\left(t\right)+n_{2}\left(t\right) \end{array}\right.$$

In Equation (Equation 5), $x_{1}\left(t\right)={\left[x_{11}\left(t\right),\ldots ,x_{1(\alpha N+1)}\left(t\right)\right]}^{T}$ and $x_{2}\left(t\right)={\left[x_{21}\left(t\right),\ldots ,x_{2(\alpha M+1)}\left(t\right)\right]}^{T}$ are the vector form of signal received at time *t*. $n_{1}\left(t\right)={\left[n_{11}\left(t\right),\ldots ,n_{1(\alpha N+1)}\left(t\right)\right]}^{T}$ and $n_{2}\left(t\right)={\left[n_{21}\left(t\right),\ldots ,n_{2(\alpha M+1)}\left(t\right)\right]}^{T}$ are the corresponding noise at time *t*. $s\left(t\right)={\left[s_{1}\left(t\right),\ldots ,s_{K}\left(t\right)\right]}^{T}$ is the signal vector at time *t*. $A_{1}\left(\theta \right)=\left[a_{11}\left(\theta \right)\;\ldots \;a_{1K}\left(\theta \right)\right]$ and $A_{2}\left(\theta \right)=\left[a_{21}\left(\theta \right)\;\ldots \;a_{2K}\left(\theta \right)\right]$ are the steering matrices of two subarrays. $a_{1k}\left(\theta \right)={\left[1\;e^{-j2\pi Mdsin\theta_{k}/\lambda }\;\ldots \;e^{-j2\pi \alpha NMdsin\theta_{k}/\lambda }\right]}^{T}$ and $a_{2k}\left(\theta \right)={\left[1\;e^{-j2\pi Ndsin\theta_{k}/\lambda }\;\ldots \;e^{-j2\pi \alpha MNdsin\theta_{k}/\lambda }\right]}^{T}$ are the steering vector, k=1,…,K.

The first sensor of the two subarrays is shared at the coordinate origin as reference sensor. The relationship between the sensors in two subarrays and the reference sensor is shown in Figure 3. There is a wave-path difference between the two sensors in each subarray. The wave-path difference between the *n*th sensor and the reference sensor of subarray one is $\tau_{n}=nMdsin\theta_{k}/c,n=1,\ldots ,\alpha N$, and the wave-path difference between the *m*th sensor and the reference sensor of subarray two is $\tau_{m}=mNdsin\theta_{k}/c,m=1,\ldots ,\alpha M$. Different wave-path differences lead to different phase lags
(6)$$\left\{\begin{array}{c} \varphi_{n}=e^{-j\omega \tau_{n}}=e^{-j\omega nMdsin\theta_{k}/c}=e^{-j2\pi fnMdsin\theta_{k}/\left(\lambda f_{0}\right)}\\ \varphi_{m}=e^{-j\omega \tau_{m}}=e^{-j\omega mNdsin\theta_{k}/c}=e^{-j2\pi fmNdsin\theta_{k}/\left(\lambda f_{0}\right)} \end{array}\right.$$

In Equation (Equation 6), $f_{0}$ is the center frequency of the signal, and because of its narrowband property, $f\approx f_{0}$. Then the phase lags $\varphi_{n}$ and $\varphi_{m}$ can be expressed as
(7)$$\left\{\begin{array}{c} \varphi_{n}=e^{-j2\pi nMdsin\theta_{k}/\lambda }\\ \varphi_{m}=e^{-j2\pi Mndsin\theta_{k}/\lambda } \end{array}\right.$$

From Equation (Equation 7), when the phase lags between two sensors are known, the DOA of the signals can be estimated.

In beamforming processing flow, in order to receive the signals from direction $\theta_{k},k=1,\ldots K$ and suppress signals from other directions, a weight vector w is needed to make the main beam pointing to the direction of $\theta_{k}$. Then the weighted outputs of the two subarrays are
(8)$$\left\{\begin{array}{c} y_{1}\left(t\right)=\langle w_{1},x_{1}\left(t\right)\rangle \\ y_{2}\left(t\right)=\langle w_{2},x_{2}\left(t\right)\rangle \end{array}\right.$$

In Equation (Equation 8), $y_{1}\left(t\right)$ is the weighted outputs of subarray one, and $y_{2}\left(t\right)$ is the weighted outputs of subarray two. When the ideal weight vectors $w_{1}$ and $w_{2}$ are fixed, the power $P_{1}\left(w_{1}\right)$ and $P_{2}\left(w_{2}\right)$ of the outputs in direction of $\theta_{k}$ can be calculated:(9)$$\left\{\begin{array}{c} P_{1}\left(w_{1}\right)=E[|y_{1}\left(t\right){|}^{2}]=E\left[y_{1}\left(t\right)y_{1}^{H}\left(t\right)\right]=E\left[w_{1}^{H}x_{1}\left(t\right){\left(w_{1}^{H}x_{1}\left(t\right)\right)}^{H}\right]=w_{1}^{H}R_{1x}w_{1}\\ P_{2}\left(w_{2}\right)=E[|y_{2}\left(t\right){|}^{2}]=E\left[y_{2}\left(t\right)y_{2}^{H}\left(t\right)\right]=E\left[w_{2}^{H}x_{2}\left(t\right){\left(w_{2}^{H}x_{2}\left(t\right)\right)}^{H}\right]=w_{2}^{H}R_{2x}w_{2} \end{array}\right.$$

In Equation (Equation 9), $R_{1x}$ is the covariance matrix of $x_{1}\left(t\right)$ in size of M×M, and $R_{2x}$ is the covariance matrix of $x_{2}\left(t\right)$ in size of N×N. Because $w_{1}$ and $w_{2}$ are the weight vectors pointed to direction $\theta_{k}$, then Equation (Equation 9) can be expressed as
(10)$$\left\{\begin{array}{c} P_{1}\left(\theta_{k}\right)=w_{1}^{H}\left(\theta_{k}\right)R_{1x}w_{1}\left(\theta_{k}\right)\\ P_{2}\left(\theta_{k}\right)=w_{2}^{H}\left(\theta_{k}\right)R_{2x}w_{2}\left(\theta_{k}\right) \end{array}\right.$$

For Equation (Equation 10), in fact we do not know where the real $\theta_{k}$ is, and we wish to estimate it. We can search within the range θ∈(−π/2,π/2) in order to get the power spectrum:(11)$$\left\{\begin{array}{c} P_{1}\left(\theta \right)=w_{1}^{H}\left(\theta \right)R_{1x}w_{1}\left(\theta \right)\\ P_{2}\left(\theta \right)=w_{2}^{H}\left(\theta \right)R_{2x}w_{2}\left(\theta \right) \end{array}\right.$$

In Equation (Equation 11), the CBF takes $w_{1}\left(\theta \right)=a_{1}\left(\theta_{k}\right)$ and $w_{2}\left(\theta \right)=a_{2}\left(\theta_{k}\right)$. Then the spatial power spectra of the two subarrays are obtained:(12)$$\left\{\begin{array}{c} P_{1}\left(\theta \right)=a_{1}^{H}\left(\theta_{k}\right)R_{1x}a_{1}\left(\theta_{k}\right)\\ P_{2}\left(\theta \right)=a_{2}^{H}\left(\theta_{k}\right)R_{2x}a_{2}\left(\theta_{k}\right) \end{array}\right.$$

For Equation (Equation 12), in real-world scenarios, the finite sample estimates of the array covariance matrix $R_{1x}$ and $R_{2x}$ are formed:(13)$$\left\{\begin{array}{c} R_{1x}=\frac{1}{p}\sum_{z=1}^{p}\left[x_{1}\left(z\right)x_{1}^{H}\left(z\right)\right]\\ R_{2x}=\frac{1}{p}\sum_{z=1}^{p}\left[x_{2}\left(z\right)x_{2}^{H}\left(z\right)\right] \end{array}\right.$$

In Equation (Equation 13), *p* is the snapshot index, $x_{1}\left(z\right)$ and $x_{2}\left(z\right)$ are the finite discrete samples, z=1…p. Therefore, the spatial spectra of the two subarrays, $P_{1}\left(\theta \right)$ and $P_{2}\left(\theta \right)$, can be obtained. However, due to the sparse spatial sampling, the spectra of the two subarrays are severely affected by grating lobes, which will influence the results of DOA estimation.

## 3. Grating Lobes Suppression

### 3.1. Single-Target Case

Let us analyze the co-prime properties of subarrays and the phase difference of the adjacent sensors, Δφ, in each subarray:(14)$$\left\{\begin{array}{c} \Delta \varphi =\;\mathrm{mod}\;(2\pi \Delta D\sin\theta /\lambda ,2\pi )\\ \Delta \varphi +2k\pi =2\pi \Delta D\sin\theta /\lambda \end{array}\right.$$

In Equation (Equation 14), mod is the modulus operator, ΔD=Qd, and *Q* is the co-prime factor *M* or *N*. *k* is an integer, and k∈[−Δφ/2π,D/λ−Δφ/2π] since θ∈(−π/2,π/2) and 0≤sinθ≤1. For a true signal direction $\theta_{t}$ and a grating lobe direction $\theta_{g}$, the relationship between their directions can be derived from Equation (Equation 14). Equation (Equation 15) shows the possible intervals between true signal directions and grating lobe directions of two subarrays,
(15)$$\left\{\begin{array}{c} \sin\theta_{t}-\sin\theta_{g}=2R_{1}/N\\ \sin\theta_{t}-\sin\theta_{g}=2R_{2}/M \end{array}\right.$$

In Equation (Equation 15), $R_{1}$ and $R_{2}$ are both integers, $R_{1}\in \left(-N,N\right)$, $R_{2}\in \left(-M,M\right)$, $\sin\theta_{t}\in \left[-1,1\right]$, $\sin\theta_{g}\in \left[-1,1\right]$. Therefore, we have $R_{1}/N=R_{2}/M$. Because of the co-prime properties of (M,N), there will be only one pair of $\left(R_{1},R_{2}\right)$ to get the true peak of the spatial spectrum, except when $R_{1}=R_{2}=0$, [24]. Hence, the beam-domain outputs of the two subarrays can be processed by the product processor to get a common peak, as is shown in Figure 4 [21].

In each bearing, the beam outputs of different subarrays are correlated and conjugated, i.e.,
(16)$$\left\{\begin{array}{c} P_{1,2}\left(\theta \right)=y_{1}\left(\theta \right){y_{2}}^{H}\left(\theta \right)\\ P_{2,1}\left(\theta \right)=y_{2}\left(\theta \right){y_{1}}^{H}\left(\theta \right) \end{array}\right.$$

In Equation (Equation 16), $y_{1}\left(\theta \right)$ and $y_{2}\left(\theta \right)$ are the beam outputs of subarrays. Take the minimum of $|P_{1,2}\left(\theta \right)|$ and $|P_{2,1}\left(\theta \right)|$, i.e.,
(17)$$P\left(\theta \right)=\min\left(\left|P_{1,2}\left(\theta \right)\right|,\left|P_{2,1}\left(\theta \right)\right|\right)$$

In Equation (Equation 17), $P_{1,2}\left(\theta \right)$ and $P_{2,1}\left(\theta \right)$ are the conjugate correlation spatial pseudo spectrum. The highest peaks are selected on the spectra that are associated with the target directions. Then the directions are obtained, which can be represented as $\left(\theta_{1},\theta_{2},\ldots ,\theta_{m}\right)$, where *m* is a positive integer. When there is only one target present, i.e., m=1, there will be only one common spectral peak in the true signal direction $\theta_{t}$ according to the deduction after Equation (Equation 14). The grating lobes can be completely suppressed in single-target scenarios with the methods in this subsection.

### 3.2. Multiple-Targets Case

For multi-targets, in addition to the true target direction, different targets in the spatial spectra of the two subarrays may generate grating lobes in the same direction, so the method for a single-target case will not be valid in some special cases. Assume that there are two targets, with target one in direction $\theta_{1}$ and target two in direction $\theta_{2}$. Two subarrays are applied to carry out beamforming simultaneously, and the grating lobes generated are related to the target directions. Therefore, the spectrum peaks obtained by subarray one include the true target directions $\theta_{1}$, $\theta_{2}$, and a group of grating lobe directions $\theta_{1g1},\ldots ,\theta_{1gu}$ (*u* is a positive integer). Similarly, the spectrum peaks obtained by subarray two include the true target directions $\theta_{1}$, $\theta_{2}$, and another group of grating lobe directions $\theta_{2g1},\theta_{2g2},\ldots ,\theta_{2gv}$ (*v* is a positive integer). When the two groups of grating lobes overlap, i.e., $\theta_{1gu}=\theta_{2gv}=\theta_{s}$, there will be a spurious peak in the direction $\theta_{s}$, which cannot be eliminated. For a spurious peak in the direction $\theta_{s}$, the following conditions are satisfied:(18)$$\left\{\begin{array}{c} \sin\theta_{s}=\sin\theta_{1}-2R_{3}/N\\ \sin\theta_{s}=\sin\theta_{2}-2R_{4}/M \end{array}\right.$$

In Equation (Equation 18), $R_{3}$ and $R_{4}$ are also integers, $R_{3}\in \left(-N,N\right)$, $R_{4}\in \left(-M,M\right)$, $\sin\theta_{1}\in \left[-1,1\right]$, $\sin\theta_{2}\in \left[-1,1\right]$, $\sin\theta_{s}\in \left[-1,1\right]$. When the equations in Equation (Equation 18) are valid, the spurious peak $\theta_{s}$ caused by the overlapping of the grating lobes cannot be eliminated, and the directions of spurious peaks satisfy Equation (Equation 19):(19)$$\sin\theta_{1}-\sin\theta_{2}=2R_{3}/N-2R_{4}/M$$

When there are multiple targets, this problem will still exist, and with an increase in the number of targets, the number of spurious peaks will continue increasing. The amplitudes of spurious peaks vary in spatial spectrum, and higher ones may even mask true signals. In underwater scenarios, the amplitudes and phases of different signals in different directions to the reference sensor are generally not the same as is mentioned in Section 2. Therefore, the amplitude information of beam-domain outputs and the phase information of the spatial cross-spectrum of the two subarrays can be utilized as the theoretical basis for evaluating whether there is a true signal or not. The theories will be elaborated in the next several paragraphs.

For the beam-domain output of each subarray, it contains the waveform information of the signals. In ideal scenarios, for the direction of true target $\theta_{t}$, the amplitudes of normalized outputs, $A_{1}\left(\theta_{t}\right)$ and $A_{2}\left(\theta_{t}\right)$, of the two subarrays are the same, i.e., $A_{1}\left(\theta_{t}\right)=A_{2}\left(\theta_{t}\right)$. However, for the direction of spurious peak $\theta_{s}$, the normalized outputs, $A_{1}\left(\theta_{s}\right)$ and $A_{2}\left(\theta_{s}\right)$, of the two subarrays are generally different, i.e., $A_{1}\left(\theta_{s}\right)\neq A_{2}\left(\theta_{s}\right)$. In practical scenarios, due to the ambient noise and interference, the actual beam-domain outputs, ${\hat{A}}_{1}\left(\theta_{t}\right)$ and ${\hat{A}}_{2}\left(\theta_{t}\right)$, of subarrays deviate from the ideal values $A_{1}\left(\theta_{t}\right)$ and $A_{2}\left(\theta_{t}\right)$. The amplitude information obtained in the true target direction is unequal, but the deviation is small, i.e., ${\hat{A}}_{1}\left(\theta_{t}\right)\approx {\hat{A}}_{2}\left(\theta_{t}\right)$. Therefore, the spurious peaks can be further distinguished by exploiting this feature. Fourier Transform is applied to the beam-domain output of the two subarrays:(20)$$\left\{\begin{array}{c} Y_{1}\left(\omega \right)=w_{1}\left(\theta \right)X_{1}\left(\omega \right)e^{-j\omega \varphi_{1}}+N_{1}\left(\omega \right)\\ Y_{2}\left(\omega \right)=w_{2}\left(\theta \right)X_{2}\left(\omega \right)e^{-j\omega \varphi_{2}}+N_{2}\left(\omega \right) \end{array}\right.$$

In Equation (Equation 20), it is assumed that the noise is white and not correlated with signals. Then the cross-spectrum of the two subarrays $G_{12}\left(\omega \right)$ is obtained.
(21)$$\begin{array}{cc} G_{12}\left(\omega \right) & =Y_{1}\left(\omega \right)Y_{2}^{*}\left(\omega \right)\\ \; & =w_{1}\left(\theta \right)w_{2}^{H}\left(\theta \right)G_{ss}\left(\omega \right)e^{-j\omega (\varphi_{2}-\varphi_{1})}+N_{1,2}\left(\omega \right) \end{array}$$

In Equation (Equation 21), due to the properties of noise, $N_{1,2}\left(\omega \right)=0$. The phase information of the spatial cross-spectrum is extracted, i.e., $\varphi =\varphi_{2}-\varphi_{1}$. The element spacings are different in the two co-prime subarrays; hence, the delays from element to element should be different as well if a plane wave is observed. Ideally, in the true target direction, the phase differences extracted from the beam-domain outputs of the two subarrays are the same, while in the direction of the spurious peaks, the phase information is generally different. Thus, to deal with the spurious peak problem, the phase feature can also be exploited. Therefore, there are two processing stages to get the true peaks. The inputs and outputs in each stage are shown in Figure 5.

#### 3.2.1. Amplitude Selection

By deduction of the above sections, the signal amplitude characteristics are introduced to identify the spurious peaks. In consideration of the amplitude characteristic of signals in real-world scenarios, we make an amplitude hypothesis. For the true target directions, the amplitude differences of the normalized outputs between the two subarrays are small; while for the spurious peak directions, the amplitude differences of the normalized outputs between the two subarrays are big. This is true for ideal sensors, but in practice each sensor has its own gain, which is direction dependent. In this paper, the sensors are assumed to have ideal omnidirectional gain patterns. The outputs of subarrays are normalized to the reference sensor, which will only eliminate the noise. As a matter of fact, the possibility that two sources with the same amplitude, phase, and distance appear in different directions is low [29]. Therefore, this case will not be included in this paper. Because of this, there will be an amplitude threshold $G_{1}$ to evaluate the peaks $\left(\theta_{1},\theta_{2},\ldots ,\theta_{m}\right)$ from the product selection. The directions that cannot meet the threshold will be eliminated, as is shown in Figure 6.

The normalized beam-domain outputs in the directions of $\left(\theta_{1},\theta_{2},\ldots ,\theta_{m}\right)$ are
(22)$$\left\{\begin{array}{c} y_{1}\left(\theta_{j}\right)=w_{1}^{H}\left(\theta_{j}\right)X_{1}/\left(\alpha N+1\right)\\ y_{2}\left(\theta_{j}\right)=w_{2}^{H}\left(\theta_{j}\right)X_{2}/\left(\alpha M+1\right) \end{array}\right.\;j=1,\ldots ,m$$

Then the amplitude information $A_{1}\left(\theta_{j}\right)$ and $A_{2}\left(\theta_{j}\right)$ can be obtained from the outputs $y_{1}$ and $y_{2}$ by
(23)$$\left\{\begin{array}{c} y_{1}\left(\theta_{j}\right)=A_{1}\left(\theta_{j}\right)e^{j\phi_{1}\left(\theta_{j}\right)}\\ y_{2}\left(\theta_{j}\right)=A_{2}\left(\theta_{j}\right)e^{j\phi_{2}\left(\theta_{j}\right)} \end{array}\right.\;j=1,\ldots ,m$$

The amplitude ratio $\rho (\theta_{j})$ of the beam-domain output of the two subarrays can be calculated, i.e.,
(24)$$\rho \left(\theta_{j}\right)=\frac{\min(|A_{1}\left(\theta_{j}\right)|,|A_{2}\left(\theta_{j}\right)|)}{\max(|A_{1}\left(\theta_{j}\right)|,|A_{2}\left(\theta_{j}\right)|)}\;j=1,\ldots ,m$$

In Equation (Equation 24), $\rho (\theta_{j})$ is the amplitude ratio with ideal value $\rho {\left(\theta_{j}\right)}_{ideal}=1$. In the real world, there can be an amplitude threshold $G_{1}$ that is a little bit less than 1. The spurious peaks will be eliminated when the amplitude ratio $\rho (\theta_{j})$ is lower than $G_{1}$. With the amplitude selection, true directions $\left(\theta_{1},\theta_{2},\ldots ,\theta_{n}\right)$, n≤m can be found.

#### 3.2.2. Phase Selection

After the amplitude selection, there may still be some spurious peaks that satisfy the amplitude threshold, and the phase information of the spatial cross-spectrum can be exploited. In consideration of the phase characteristics of signals in real-world scenarios, we make a phase hypothesis. For the true target directions, the phase differences extracted from the beam-domain outputs of the two subarrays are small; while for the spurious peak directions, the phase differences extracted from the beam-domain outputs of the two subarrays are big. The phase differences extracted above correspond to the spatial cross-spectra of the two subarrays. Since the sensor displacements are different, the delays will be different in the two co-prime array components. Take two targets in directions of $0^{{}^{\circ}}$ and $10^{{}^{\circ}}$ for example, the phase is extracted in Table 1, and Figure 7 also illustrates the case. Because of this, there can be a phase threshold $G_{2}$ to evaluate the peaks $\left(\theta_{1},\theta_{2},\ldots ,\theta_{n}\right)$ from the amplitude selection. The directions that cannot meet the threshold will be eliminated, as is shown in Figure 8. The normalized beam-domain outputs in the directions of $\left(\theta_{1},\theta_{2},\ldots ,\theta_{n}\right)$ are
(25)$$\left\{\begin{array}{c} y_{1}\left(\theta_{j}\right)=w_{1}^{H}\left(\theta_{j}\right)X_{1}/\left(\alpha N+1\right)\\ y_{2}\left(\theta_{j}\right)=w_{2}^{H}\left(\theta_{j}\right)X_{2}/\left(\alpha M+1\right) \end{array}\right.\;j=1,\ldots ,n$$

The phase information $\phi_{1}\left(\theta_{j}\right)$ and $\phi_{2}\left(\theta_{j}\right)$ can be obtained from the outputs $y_{1}\left(\theta_{j}\right)$ and $y_{2}\left(\theta_{j}\right)$ by
(26)$$\left\{\begin{array}{c} y_{1}\left(\theta_{j}\right)=A_{1}\left(\theta_{j}\right)e^{\phi_{1}\left(\theta_{j}\right)}\\ y_{2}\left(\theta_{j}\right)=A_{2}\left(\theta_{j}\right)e^{\phi_{1}\left(\theta_{j}\right)} \end{array}\right.\;j=1,\ldots ,n$$

The phase difference $\Delta \phi (\theta_{j})$ of the beam-domain outputs of the two subarrays is
(27)$$\Delta \phi \left(\theta_{j}\right)=\left|\phi_{1}\left(\theta_{j}\right)-\phi_{2}\left(\theta_{j}\right)\right|$$

In Equation (Equation 27), $\Delta \phi (\theta_{j})$ is a phase difference corresponding to the phase information of the spatial cross-spectrum of the two subarrays with ideal value $\Delta \phi {\left(\theta_{j}\right)}_{ideal}=0$, which can be derived from Equation (Equation 21). In the real world, there will be a phase threshold $G_{2}$ a little bit more than 0. The spurious peaks can be further eliminated when $\Delta \phi (\theta_{j})$ is higher than $G_{2}$. In this way, true target directions $\left(\theta_{1},\theta_{2},\ldots ,\theta_{k}\right)$, k≤n can be obtained in phase selection.

### 3.3. Processing Flow

In this paper, co-prime array geometry is employed. The processing flow is shown in Figure 9. To start with, the subarray CBF is utilized to obtain the initial directions. Then, the beam-domain outputs of subarrays in each bearing are conjugated and multiplied to obtain the peaks of the correlation spectrum with the ambiguity being eliminated. Based on this, the amplitude information from two subarrays is exploited to further discriminate the spurious peaks with the amplitude threshold. After that, the phase information is exploited to pick out spurious peaks that have not been distinguished by the amplitude selection stage. Finally, all spurious peaks in the spatial spectrum can be identified, and the ambiguity will be eliminated. Following the steps in the processing flow, the true directions of targets are more deterministic, which is beneficial to interesting fields such as UUV sonar arrays [31].

This paper focuses on the ambiguity elimination problem rather than an underwater detection problem, so it is assumed that the SNR is enough to execute the processing flow. The amplitude and phase information can be estimated in a reasonable range.

## 4. Simulation and Discussion

The parameters of CA are chosen first: The co-prime pair is (M,N)=(3,4), and the extension factor is α=3. Thus, the sensor number of the two subarrays is $\left(M_{1},M_{2}\right)=\left(13,10\right)$. The total number of sensors generated by the subarrays is $M_{SUM}=19$, and the sensor number of ULA with the same aperture is $M_{ULA}=37$. The resolution of a certain direction is directly related to the changing rate of the steering vector near the direction, which can be given in Equation (Equation 28):(28)$$D\left(\theta \right)=∥\frac{da(\theta )}{d\theta }∥\propto ∥\frac{d\tau }{d\theta }∥$$

The Rayleigh Criterion for the ULA is 3.1°. Assume that all the sensors are identical, reciprocal, and omnidirectional. The center frequency of signals is set at $f_{0}=1.5\;\mathrm{kHz}$, and the sampling frequency is $f_{s}=15\;\mathrm{kHz}$. The underwater sound velocity is c=1500m/s, and the sensor spacings in subarrays are $\Delta d_{1}=1.5\;m\;\left(1.5\;\lambda \right)$ and $\Delta d_{2}=2.0\;m\;\left(2.0\;\lambda \right)$, respectively. CBF is adopted, and the bearing range is θ∈(−π/2,π/2).

### 4.1. Processing Method Simulation

Assume a scenario that K=2 far-field point targets are simulated. The rationality and correctness of the algorithm are verified by changing the parameters of targets. The amplitude threshold and the phase threshold are set at $\left(G_{1},G_{2}\right)=\left(0.95,4\right)$. The solid lines represent the true target directions, and the dotted lines represent the spurious peak directions.

#### 4.1.1. Product Processing

The directions of two targets are $\left(\theta_{1},\theta_{2}\right)=\left(0^{{}^{\circ}},12.5{}^{\circ}\right)$. The SNR level is $\left(SNR_{1},SNR_{2}\right)=\left(0,0\right)$, and the phases of targets are set randomly. The number of snapshots is 1000, and the simulation result is shown in Figure 10a.

In this situation, the spurious peaks can be eliminated by product processing, but when the direction of targets changes to $\left(\theta_{1},\theta_{2}\right)=\left(0{}^{\circ},10{}^{\circ}\right)$, the spurious peaks cannot be eliminated, as is shown in Figure 10b. Since the main beamwidth of the beamformer is about 3.1°, the second target is located out of the beam range.

#### 4.1.2. Amplitude Selection

The directions of two targets are $\left(\theta_{1},\theta_{2}\right)=\left(0{}^{\circ},10{}^{\circ}\right)$. The SNR level is $\left(SNR_{1},SNR_{2}\right)=\left(0,5\right)$, and the phases of targets are set randomly. The number of snapshots is 1000, and the simulation result is shown in Figure 11a.

In this situation, the spurious peaks cannot be distinguished by product processing. With the amplitude selection, the spurious peaks can be identified. However, when two targets have similar SNRs, for example when $\left(SNR_{1},SNR_{2}\right)=\left(0,0\right)$, the amplitude selection is not as satisfactory, as is shown in Figure 11b. Additionally, individual gain patterns make this even more difficult when the sensors are not directionless.

#### 4.1.3. Phase Selection

The directions of two targets are $\left(\theta_{1},\theta_{2}\right)=\left(0{}^{\circ},10{}^{\circ}\right)$. The SNR level is $\left(SNR_{1},SNR_{2}\right)=\left(0,0\right)$, and the phases of targets are set randomly. The number of snapshots is 1000, and the simulation result is shown in Figure 12a.

In this situation, the spurious peaks cannot be distinguished by amplitude selection. With phase selection, the spurious peaks can be identified, as is shown in Figure 12b. However, when a true signal overlaps with a spurious peak, or two signals overlap with each other, the processing does not work well enough. The resolution limit of the array is 3.1°, and the sources are in the same direction (within the main beam). The mainlobe interference elimination, blind source separation, and independent component analysis can be exploited to solve this problem in further work.

#### 4.1.4. Comparison

After the processing flow is simulated, the spatial spectrum obtained by CBF of whole array, the spatial spectrum obtained by MUSIC of whole array, the proposed method with co-prime array in this paper, and the half-wavelength ULA with the same aperture are compared.

As is shown in Figure 13, comparing with the CBF method applied to the whole array, the proposed method in this paper can effectively distinguish spurious peaks and accurately estimate the true direction of the targets. The MUISC algorithm is also applied to the whole array under the condition that the number of signals is known, and the spurious problem still exists, which is similar to the situation when CBF is applied to the whole array. Comparing with the half-wavelength ULA with same aperture, the spatial spectral peaks estimated by the proposed method are close to the width of the ULA results, which can meet the estimation requirements.

### 4.2. Probability of Success (PoS)

#### 4.2.1. The Selection of Thresholds

The selection thresholds $G_{1}$ and $G_{2}$ are highly related to false alarms and underreporting, which has considerable influence on the PoS of the proposed method in this paper. Before discussing the PoS for the two targets case and the multiple targets case, the selection of thresholds should be simulated. As is stated in Section 3, the amplitude threshold $G_{1}$ is a little bit less than 1, and the phase threshold $G_{2}$ is a little bit more than 0, theoretically.

The conditions of simulation remain unchanged, and the parameters of targets remain the same as in the last section. In the processing flow, the amplitude selection is achieved before the phase selection, and the relationship between PoS and $G_{1}$ is simulated. The range of amplitude threshold $G_{1}$ is adjusted from 0.8 to 1 with the interval 0.02. For each $G_{1}$, 500 Monte Carlo experiments were conducted, and the PoS versus $G_{1}$ is obtained. As is shown in Figure 14a, the PoS goes steadily up with the increase of $G_{1}$. The amplitude threshold $G_{1}$ gradually eliminates the angle which is not qualified to it. When $G_{1}=0.94$, the PoS reaches the max with PoS = 81.2% and the value of $G_{1}$ near 0.95 is recommended to be the amplitude threshold. In this range, the amplitude selection can guarantee at least PoS = 70%. Then, the PoS goes down until PoS = 0 when $G_{1}=1$. In this situation, the threshold is too high and approaches the ideal circumstance, which is not a good range for the selection of $G_{1}$.

After selection of the amplitude threshold $G_{1}$, the selection of phase threshold is simulated. The conditions of simulation remain unchanged, and the parameters of targets remain the same. The amplitude $G_{1}$ is set at 0.95, as a reliable value from the simulation result. The range of phase threshold $G_{2}$ is adjusted from 0 to 10 with the interval 0.5. For each $G_{2}$, 500 Monte Carlo experiments were conducted, and the PoS with the corresponding $G_{2}$ is obtained. As is shown in Figure 14b, the PoS increases sharply when $G_{2}\leq 3$. When $G_{2}$ is in the range of 3.5 to 4.5, the PoS reaches the top with PoS = 98%, and the values near here are recommended to be the phase threshold. Then, the PoS decreases slowly, for $G_{1}$ has provided enough guarantee for the phase selection. The simulations are based on two targets here and in practice, and the two thresholds can be set artificially according to empirical experience or adaptive methods.

#### 4.2.2. PoS for Two Targets

In order to verify the ambiguity elimination effect when the directions of targets change, the SNR level is fixed at $\left(SNR_{1},SNR_{2}\right)=\left(0,0\right)$. The direction of target one remains unchanged $\theta_{1}=0{}^{\circ}$, and the direction of target two changes within the range $\theta_{2}\in \left(-90{}^{\circ}:90{}^{\circ}\right)$. In each bearing, 1000 Monte Carlo experiments are conducted, and the PoS versus corresponding $\theta_{2}$ is obtained.

As is shown in Figure 15, when target two is located in the direction of $\theta_{2}\in \left(-3{}^{\circ}:3{}^{\circ}\right)$, the PoS decreases significantly. This is caused by the overlap of the two signals, which has been discussed in Section 3. When $\theta_{2}\in \left(-89{}^{\circ}:90{}^{\circ}\right)$, the end-fire direction of the array, the DOA ability of the array decreases obviously. While in most directions, the PoS remains stable with little fluctuation.

#### 4.2.3. PoS for Multiple Targets

As is shown in Section 3, when the number of targets increases, the number of spurious peaks will continue increase. The simulation of PoS versus the number of targets in a certain range is carried out. The SNR of targets are fixed between 0 dB and 3 dB. The direction of target one remains unchanged at 0°, and the direction of other targets changes with different angle intervals from 3° to 15°, as is shown in Figure 16a. In each case, 1000 Monte Carlo experiments were conducted, and the PoS was obtained.

As is shown in Figure 16b, when the interval of targets is 3°, the PoS remains stable near 100% with the number of targets under 8. However, as the number of targets increases, the PoS decreases, and when the number of targets reaches 11 the method becomes invalid. This is because when more targets appear, the component of spurious peaks becomes more complex. When the amplitude and phase information extracted from this spurious peak are similar to that of true targets, the existence of targets cannot be evaluated, i.e., the spurious peak will be identified as a true peak. When the spurious peak and the true peak overlap in the same direction, the true peak would also be identified as a spurious peak. When the interval of targets increases, the number of identifiable targets decreases. Therefore, the method proposed in this paper has better performance in scenarios with fewer targets. When the scenarios become complicated, it is recommended to apply and combine more methods like independent component analysis to accomplish the estimation.

### 4.3. Statistical Performance Analysis

The root-mean-square error (RMSE) of the method proposed in this paper versus SNR and the number of snapshots are next investigated over 1000 independent Monte Carlo simulations. The RMSE of DOA estimation is calculated as
(29)$$\mathrm{RMSE}=\sqrt{\frac{1}{K\psi }\sum_{\kappa =1}^{\psi }\sum_{k=1}^{K}{\left({\hat{\theta }}_{k\kappa }-\theta_{k}\right)}^{2}},$$

In Equation (Equation 29), ψ is the number of Monte Carlo simulations, κ=1,…,ψ, *K* is the number of sources, $\theta_{k}$ is the *k*th real direction of the signal, and ${\hat{\theta }}_{k\kappa }$ is the estimated direction of the *k*th source in the κth Monte Carlo simulation, k=1,…,K.

The directions of targets are fixed at $\left(\theta_{1},\theta_{2}\right)=\left(0{}^{\circ},10{}^{\circ}\right)$. The number of snapshots is 1000, and the SNR level is adjusted from −15 dB to 0 dB. The thresholds are set at $\left(G_{1},G_{2}\right)=\left(0.95,4\right)$. The RMSE versus SNR with the proposed method is shown in Figure 17a. The RMSE appears to decrease gradually as the SNR of the signals increases. When the SNR level is low, the presence of the noise seriously affects the accuracy of DOA estimation. When the SNR level is sufficiently high, the method proposed in this paper presents the similar accuracy with little RMSE. When the SNR level reaches −9 dB, the RMSE remains at 0.

The RMSE versus snapshots is then simulated. The SNR level of targets is fixed at 0 dB, i.e., $\left(SNR_{1},SNR_{2}\right)=\left(0,0\right)$, and the number of snapshots is adjusted from 100 to 1000. As is shown in Figure 17, when the number of snapshots increases, the RMSE descends gradually. When there are sufficient snapshots, the RMSE remains stable near 0. However, compared to the RMSE versus SNR level, RMSE versus snapshots has less influence on the proposed method. From the statistical performance analysis simulation, the proposed method can guarantee the DOA estimation performance under sufficient SNR and achieve similar performance with insufficient snapshots.

### 4.4. Prospects For Future Work

The method in this paper is proposed to solve the problem of grating lobes because of the sparse spatial sampling in passive sensing applications. In practical sonar system applications, how to quickly eliminate ambiguity is a practical problem. Strong demand for wide-band signal processing and vector hydrophone array processing with sparse arrays under random interferences still exists. In this paper, the reduction of mutual coupling and relevant material effect are qualitatively discussed, which can be verified with more experiments. The threshold selection in different cases depends on the parameter setting, and the rules will be further studied.

## 5. Conclusions

CAs are gradually applied in underwater scenarios because they have significant advantages over traditional ULAs. For the same sensor number, CAs provide better accuracy and higher resolution. While for the same aperture, CAs can reduce the physical cost. Because of the expansion of inter-spacing, the mutual coupling can be reduced. Despite the advantages of CAs, the spatial spectra obtained directly from CBF can be degraded by grating lobes due to the sparse spatial sampling in passive sensing applications, which will seriously deteriorate the estimation performance. Therefore, the ambiguity elimination based on CAs is worthy of study. The ambiguity elimination in this paper is solved taking advantage of the co-prime property of subarrays in CAs. The amplitude and phase information of beam-domain outputs are exploited based on this idea.

The method proposed in this paper is applied to the CA as a pretreatment before the high-resolution DOA estimation algorithms like MUSIC, etc. In the processing flow, the subarray CBF is firstly utilized to obtain the initial direction set. Then, the beam-domain outputs of subarrays in each bearing are conjugated and multiplied to obtain the peaks of the correlation spectrum. On the basis of this, the amplitude information from two subarrays is exploited to further discriminate the spurious peaks with the amplitude threshold. After the amplitude selection, the phase information is exploited to identify spurious peaks that are not distinguished by the amplitude selection. Finally, all the spurious peaks in the spatial spectrum can be identified, and the ambiguity is eliminated.

Simulation results validated the effectiveness of the proposed method in this paper. The selection of the threshold is discussed, and the recommended values are given. The statistical performance analysis of the proposed method is studied and proofed to guarantee the performance. To summarize, the method with CA utilizes fewer sensors to estimate the DOA accurately and can eliminate the ambiguity caused by at least three targets. When the angle range of interest is given, this method performs better. In practical scenarios like underwater applications, the proposed method has great practical significance. It can effectively promote the related array signal processing algorithm in reality, save costs, and is easy to implement.

## Figures and Tables

**Figure 1 sensors-20-06058-f001:**
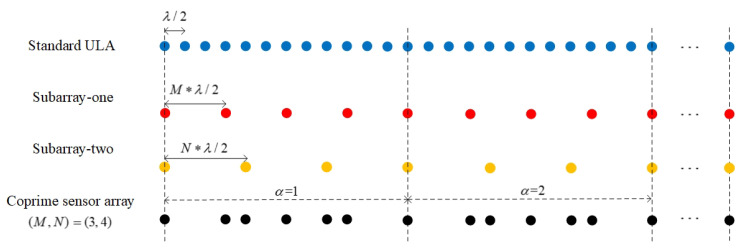
Geometry of the co-prime array.

**Figure 2 sensors-20-06058-f002:**
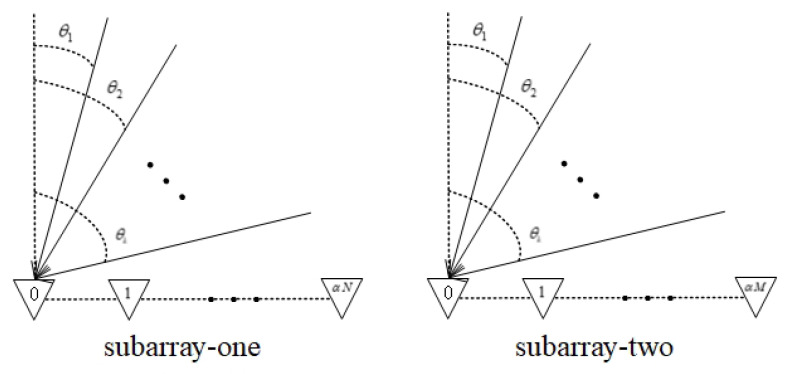
The relationship between targets and array.

**Figure 3 sensors-20-06058-f003:**
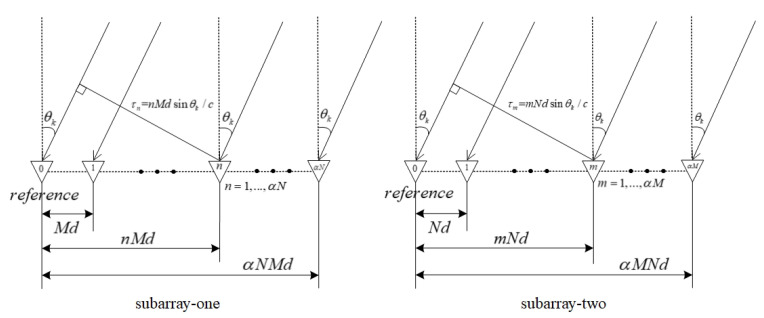
Geometric relationship between sensors in the far-field scenario.

**Figure 4 sensors-20-06058-f004:**
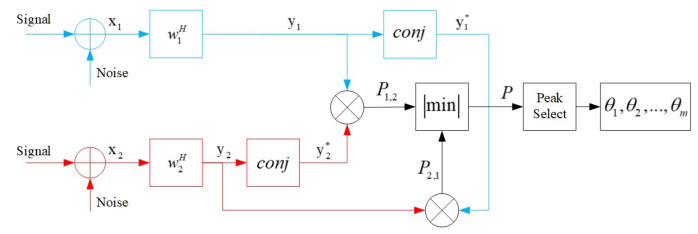
Product processor.

**Figure 5 sensors-20-06058-f005:**
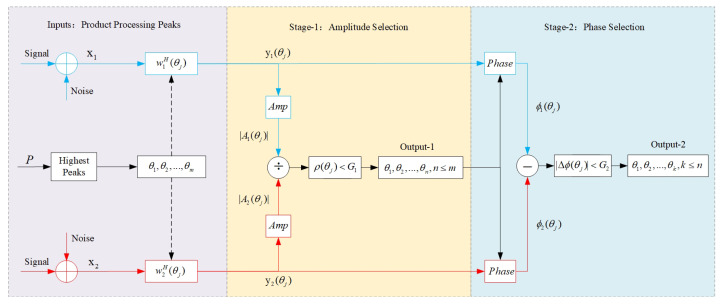
Two stages of processing.

**Figure 6 sensors-20-06058-f006:**
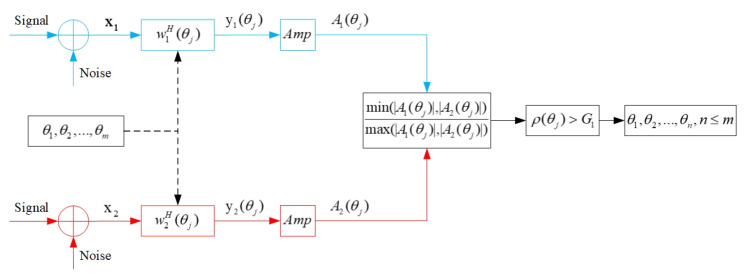
Amplitude selection.

**Figure 7 sensors-20-06058-f007:**
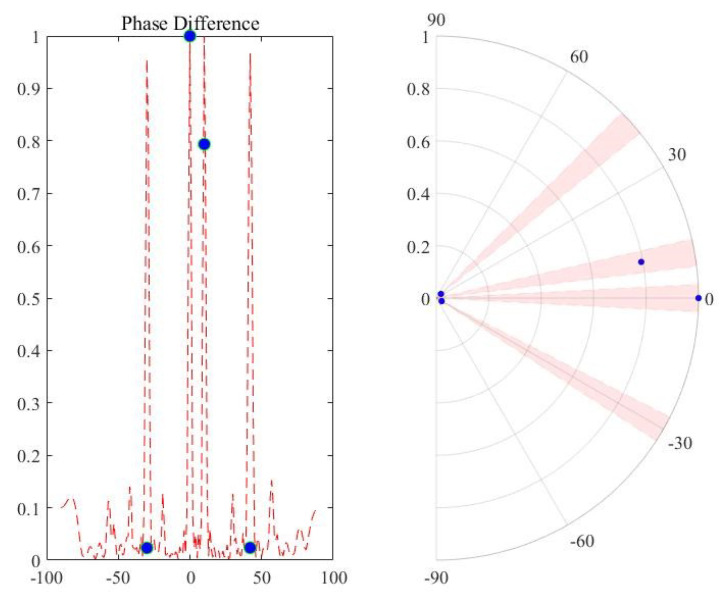
True peaks with small cross-spectrum phases.

**Figure 8 sensors-20-06058-f008:**
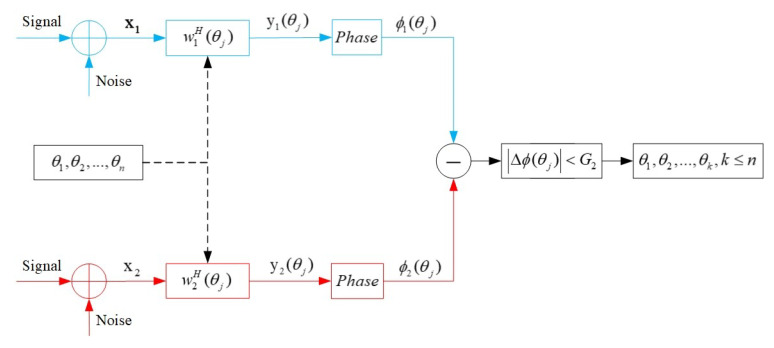
Phase selection.

**Figure 9 sensors-20-06058-f009:**
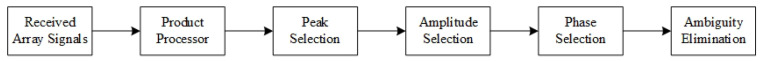
Processing flow.

**Figure 10 sensors-20-06058-f010:**
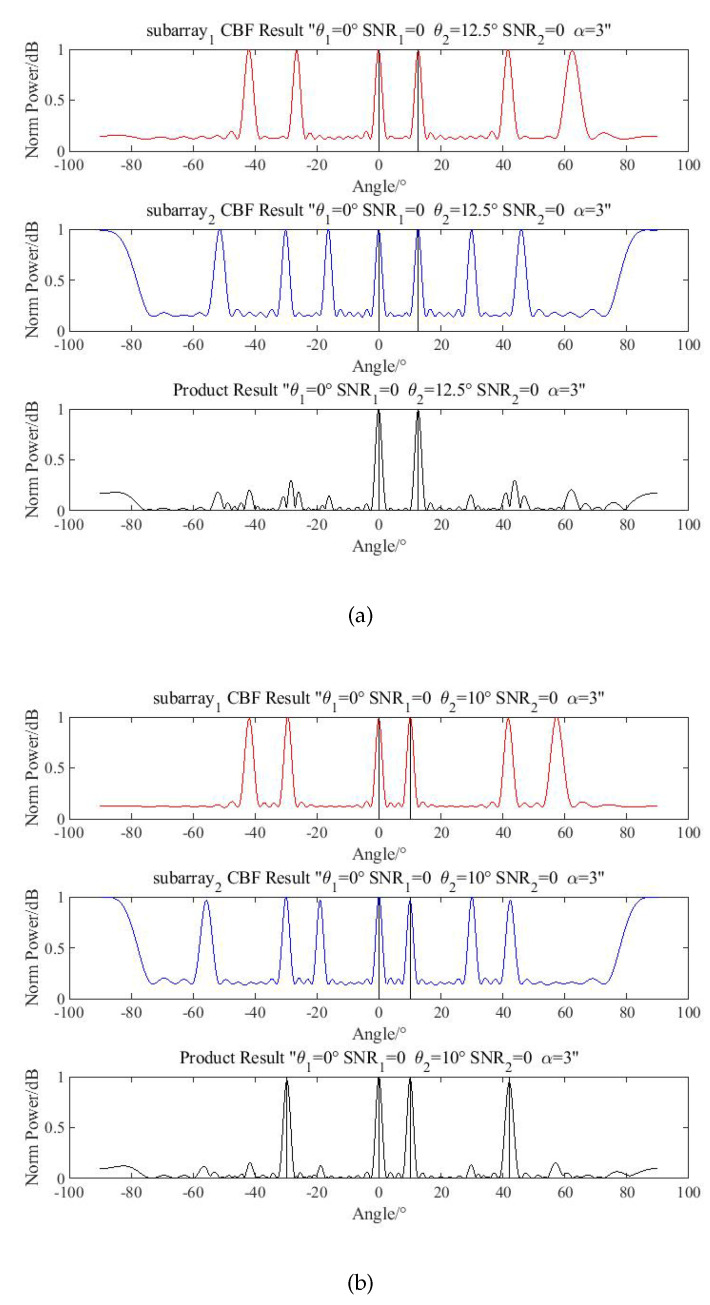
Simulation of product processing. (**a**) Eliminate successfully with product processing; (**b**) Fail to eliminate spurious peaks with product processing.

**Figure 11 sensors-20-06058-f011:**
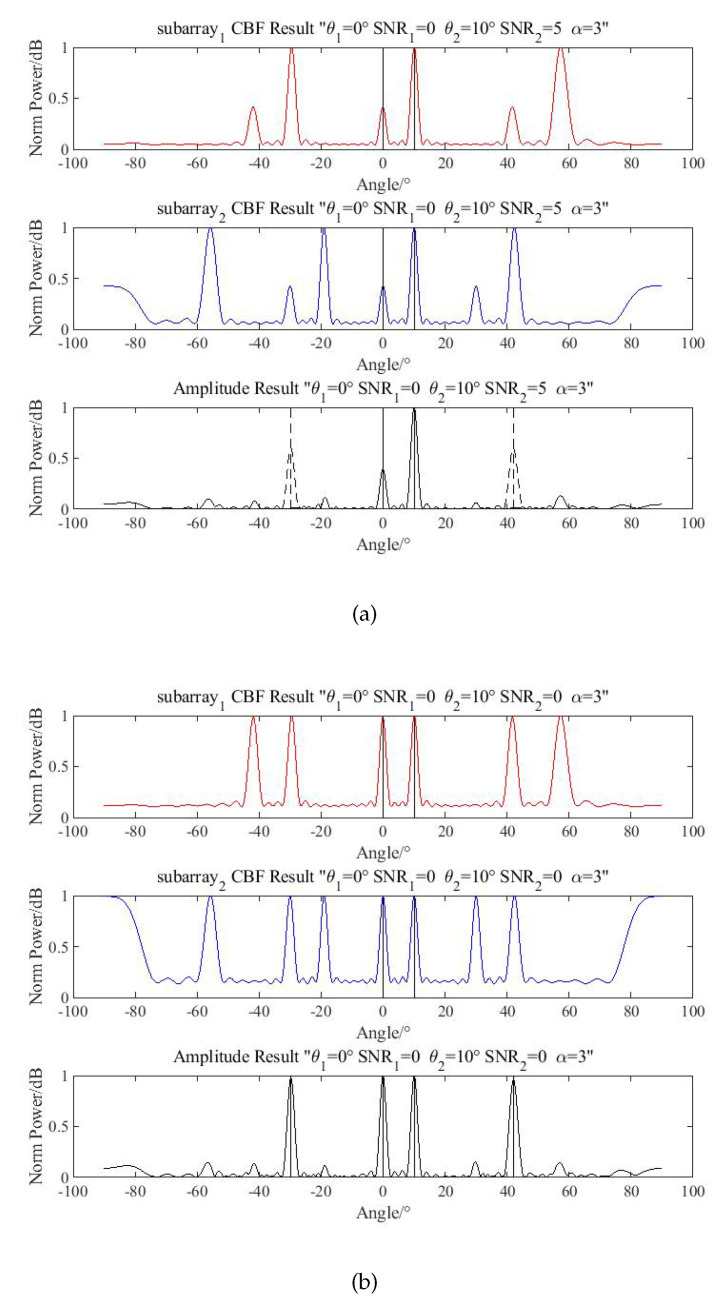
Simulation of amplitude selection. (**a**) Eliminate successfully with amplitude selection; (**b**) Fail to eliminate spurious peaks with amplitude selection.

**Figure 12 sensors-20-06058-f012:**
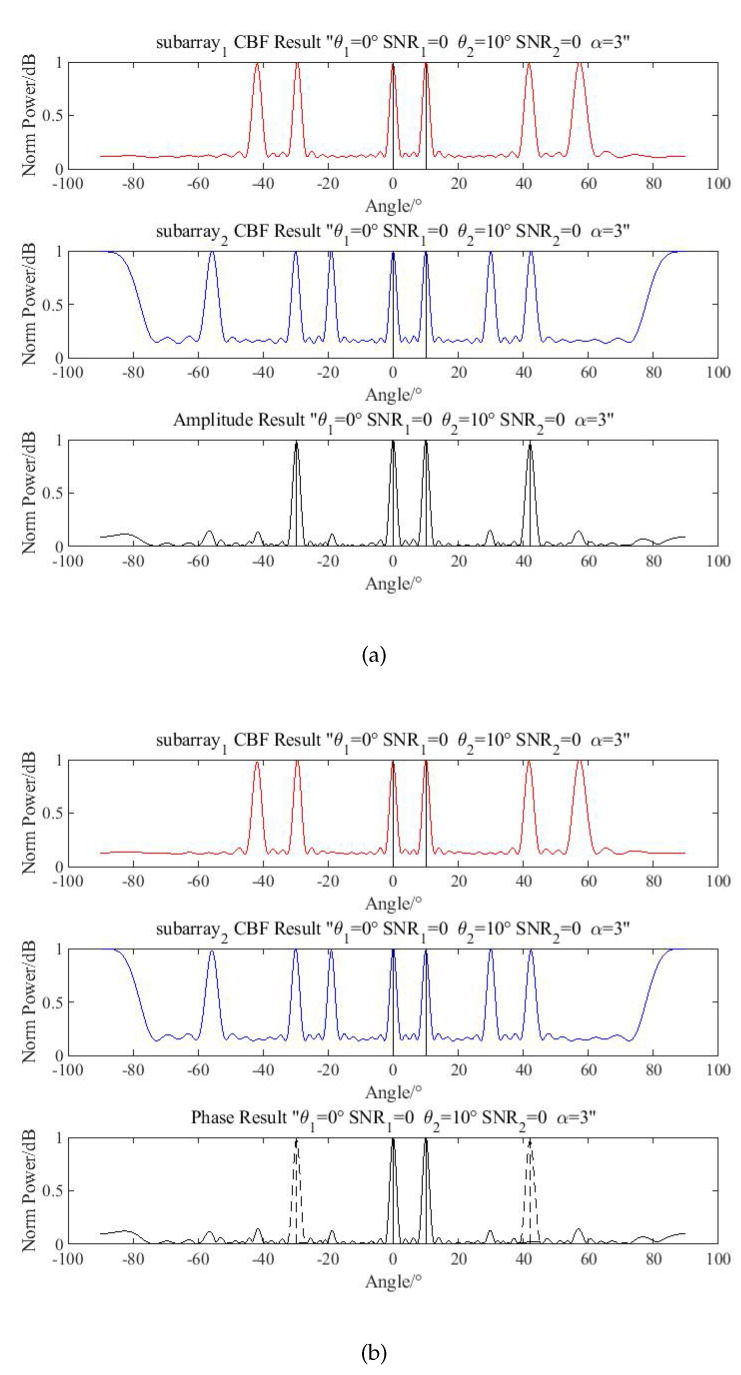
Simulation of phase selection. (**a**) Fail to eliminate spurious peaks with amplitude selection; (**b**) Eliminate successfully with phase selection.

**Figure 13 sensors-20-06058-f013:**
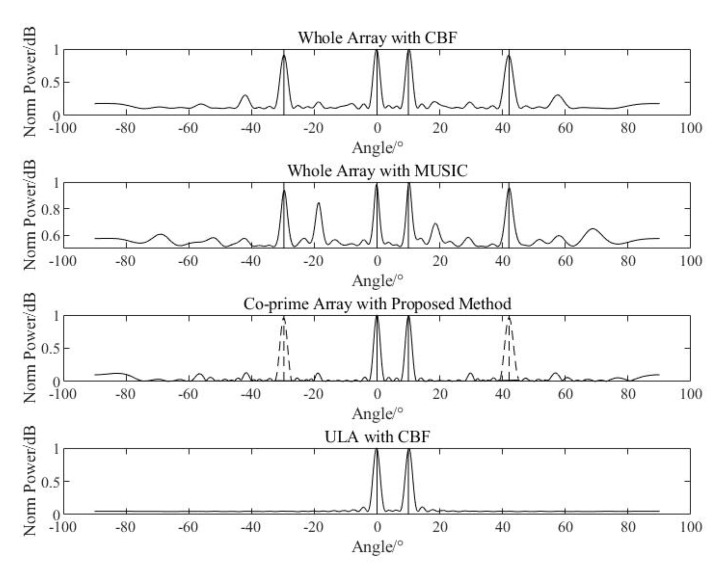
Whole-array CBF, whole-array MUSIC, co-prime array proposed method, and ULA CBF.

**Figure 14 sensors-20-06058-f014:**
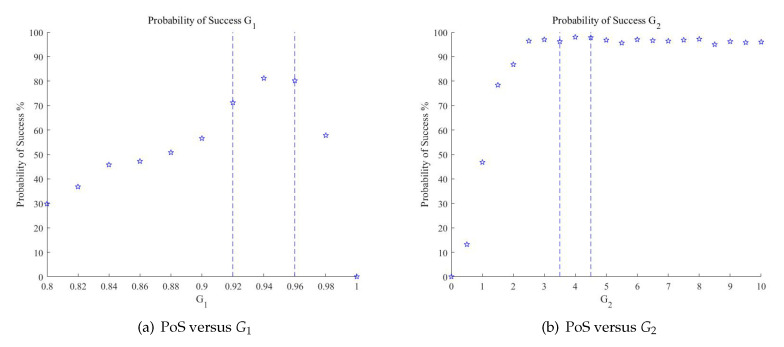
Probability of success versus $G_{1}$ and $G_{2}$.

**Figure 15 sensors-20-06058-f015:**
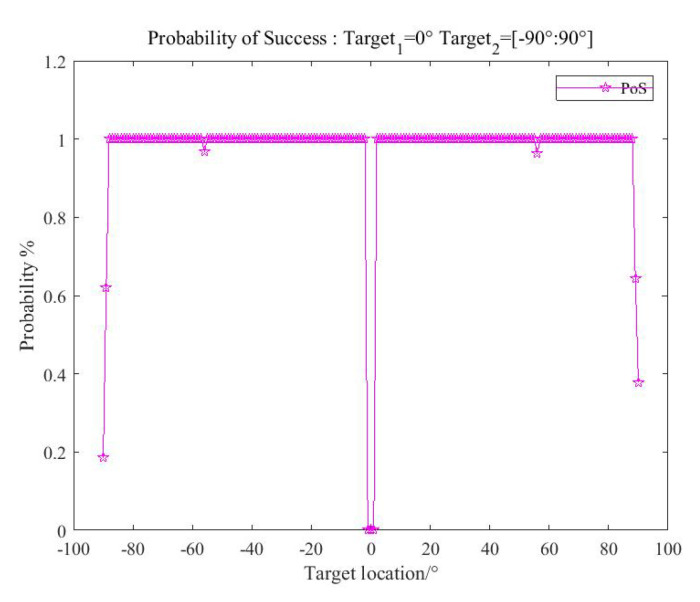
Probability of success versus target location.

**Figure 16 sensors-20-06058-f016:**
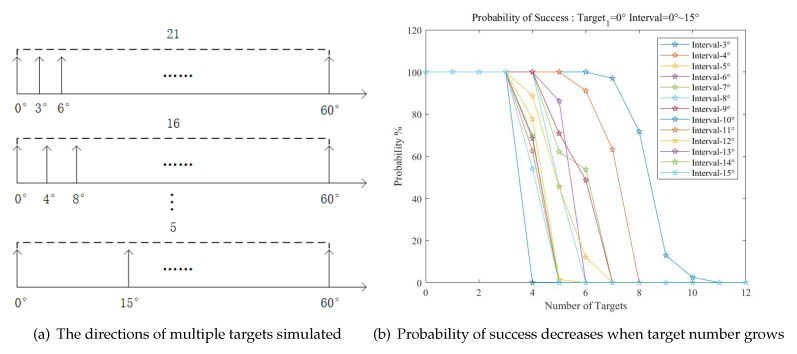
Multiple-targets simulation.

**Figure 17 sensors-20-06058-f017:**
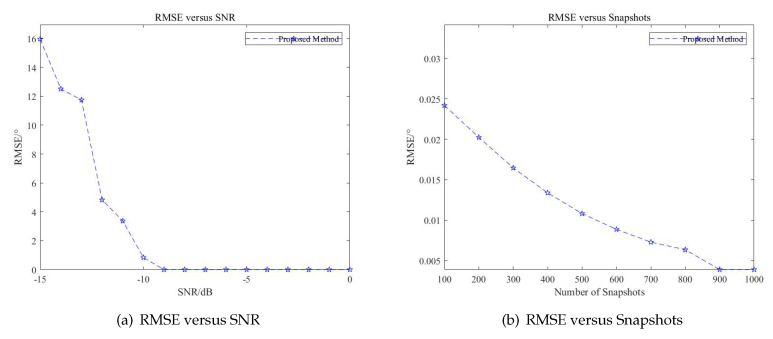
Statistical performance analysis.

**Table 1 sensors-20-06058-t001:** Simulations on cross-spectrum phases.

Directions / °	−30°	0°	10°	42°
Montecarlo 100 times	18.7452	0.4327	0.4799	18.5662
Montecarlo 1000 times	18.7620	0.4388	0.4518	18.5439

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
