# Peer review of "Underwater Ambiguity Elimination Method Based on Co-Prime Sensor Array"

_sensors, 2020, doi:10.3390/s20216058_

Round 1
Reviewer 1 Report
In this paper, the authors describe an ambiguity elimination method by using the amplitude and phase information of CA beam-domain outputs. This topic is interesting. However, the paper lacks of some important discussions, such as discussion of threshold, performance discussion with respect to SNR. Summarily, the paper should present the work comprehensively. Therefore, the authors should enhance these discussions to further verify their method.
- Sk(t) in Eq. (3) is the signal vector. Therefore, it is independent of the variable k. At this point, it suggested to be replaced by vector S(t) .
- The vector ai(θ) should be first defined in Eq. (6).
- The covariance matrix shown in Eq. (7) is not right. The authors should check this formula.
- In sections 3.2.1 and 3.2.2, the authors exploit the thresholds G1 and G2, which are very important for the authors’ method. The reviewer wanders to know how to determine both thresholds. The authors should discuss both thresholds in detail as they highly related to the false alarm and underreporting.
- Fig. 5 is not in accordant with Eq. (18) as ρ=A1/A2 in Fig. 5. The authors should check this issue.
- The simulations should be enhanced. Although the authors said that their method works in the condition of enough SNR, the authors should discuss the performance of presented method with respect to SNR. This is very important for the readers’ choice whether this method is exploited by readers. Furthermore, the method performance should also be discussed with different snapshots.
- In line 301 on page 13, 3db -> 3°. The authors should proofread the paper to avoid similar errors.
- The English in this paper should be further improved.
Author Response
On behalf of my co-authors, we really appreciate the Editor and Reviewers very much for the positive and constructive comments and suggestions on our manuscript entitled “Underwater Ambiguity Elimination Method Based on Co-prime Sensor Array” (Manuscript ID: sensors-957839). These comments are very valuable and helpful for revising and improving our paper.
We have studied the reviewers’ comments carefully and have made careful modifications to the original manuscript, which will make our best to revise the manuscript according to the comments.
Attached is our response to the comments, where, red italic letters indicate the comment, black letters indicate our reply and letters with yellow background in our revised manuscript indicates our revisions.
Please see the attachment.

Reviewer 2 Report
My impression is that the authors have developed a method of supressing spurious peaks in DOA estimation coprime ULA arrays that works in practice but mathematical foundations behind it are not clear and the data model have been included just to be there.
My main objections are about the mathematical formulation:
- Why did you choose a real-valued signal in (1)?
- In l. 93-94 the statement “the original signal parameters of sources are basically different from each other” is not clear and also it is not clear what assumptions are made in the statement that influence further development.
- In (2) \theta_k is not defined, in spite of their estimation being the objective of the paper
- The S_k(t) is defined as [s_1(t),\ldots, s_k(t)]^T, k = 1, \ldots, K. Clearly it should be [s_1(t),\ldots, s_K(t)]^T
- In Figure 2 you assume far-field scenario. This is never stated.
- In Figure 2 the distance between the sensor l and sensor 0 is denoted as d_{0l}, however in l. 106 you state it is equal to d_{\alpha N + 1}k. First, the $l$ does not appear in the indexing, second, it is not coherent with the Figure 2. \alpha N + 1 in d_{\alpha N + 1}k is just the index of the last sensor in the array.
- Consequently, the indexing in (4) is also wrong. The number of targets is K, not k. And the notation is incoherent with the Figure 2.
- The time lag \tau is again specified for the last sensor only and also the transition from 2\pi/\lambda to \omega/c is not explained.
- In (5) neither y_i nor w_i is defined. In the preceding sentences you say it is the weight, but what kind and why is it there?
- In (7) X_i is not defined, and what’s more, it is discrete-time, incoherent with (3) in which X_(t) is continuous-time. In fact (1)-(3) are all continuous time. Also n is used here as a local index, but n was already taken earlier for a different purpose.
Unclear on what is the data model, I skipped most of the section 3, but two theorems caught my eye. These are not in any way theorems! They are just unsupported statements. And even if they were, a proof would be required. In any way, these statements must be supported because they are used later and have impact on the results.
- Thresholds G_1 and G_2 appear all of a sudden and are not explained.
- In Section 3.3. you use CBF to obtain initial directions. What if they are completely wrong? After all, CBF, especially in low SNR conditions is a not very reliable method. Why not use MUSIC or other super resolution technique?
- Why do you use 15 kHz as the sampling rate? It is 10 times the f_0.
- 276 What do you mean by “hollistic”?
Some more notation problems:
- In Figure 1 you say subarray-one, while in text you say sub-array-1
- an pre-processeing
- spectrums
- thanks to (colloquial)
- AD-HOC (why capitals?)
- l. 30 the stated interest in recent years is not backed by references
- l.113 which is in size
- KHz should be kHz
- Units should be spaced from the values
- I recommend writing l-th and not lth (lloks like product of three variables).
Author Response

(The authors gave the same response as above.)

Round 2
Reviewer 1 Report
The authors well address the reviewer's comments. To some degree, the paper has been improved after revision. The reviewer suggests that the paper is accpeted for publication.